# Structured Transforms for Small-Footprint Deep Learning

**Vikas Sindhwani**      **Tara N. Sainath**      **Sanjiv Kumar**
Google, New York
{sindhwani, tsainath, sanjivk}@google.com

## Abstract

We consider the task of building compact deep learning pipelines suitable for deployment on storage and power constrained mobile devices. We propose a unified framework to learn a broad family of structured parameter matrices that are characterized by the notion of low displacement rank. Our structured transforms admit fast function and gradient evaluation, and span a rich range of parameter sharing configurations whose statistical modeling capacity can be explicitly tuned along a continuum from structured to unstructured. Experimental results show that these transforms can significantly accelerate inference and forward/backward passes during training, and offer superior accuracy-compactness-speed tradeoffs in comparison to a number of existing techniques. In keyword spotting applications in mobile speech recognition, our methods are much more effective than standard linear low-rank bottleneck layers and nearly retain the performance of state of the art models, while providing more than 3.5-fold compression.

## 1   Introduction

Non-linear vector-valued transforms of the form, $f(\mathbf{x}, \mathbf{M}) = s(\mathbf{Mx})$, where $s$ is an elementwise nonlinearity, $\mathbf{x}$ is an input vector, and $\mathbf{M}$ is an $m \times n$ matrix of parameters are building blocks of complex deep learning pipelines and non-parametric function estimators arising in randomized kernel methods [20]. When $\mathbf{M}$ is a large general dense matrix, the cost of storing $mn$ parameters and computing matrix-vector products in $O(mn)$ time can make it prohibitive to deploy such models on lightweight mobile devices and wearables where battery life is precious and storage is limited. This is particularly relevant for "always-on" mobile applications, such as continuously looking for specific keywords spoken by the user or processing a live video stream onboard a mobile robot. In such settings, the models may need to be hosted on specialized low-power digital signal processing components which are even more resource constrained than the device CPU.

A parsimonious structure typically imposed on parameter matrices is that of low-rankness [22]. If $\mathbf{M}$ is a rank $r$ matrix, with $r \ll \min(m, n)$, then it has a (non-unique) product representation of the form $\mathbf{M} = \mathbf{GH}^T$ where $\mathbf{G}, \mathbf{H}$ have only $r$ columns. Clearly, this representation reduces the storage requirements to $(mr + nr)$ parameters, and accelerates the matrix-vector multiplication time to $O(mr+nr)$ via $\mathbf{Mx} = \mathbf{G}(\mathbf{H}^T\mathbf{x})$. Another popular structure is that of sparsity [6] typically imposed during optimization via zero-inducing $l_0$ or $l_1$ regularizers. Other techniques include freezing $\mathbf{M}$ to be a random matrix as motivated via approximations to kernel functions [20], storing $\mathbf{M}$ in low fixed-precision formats [7, 24], using specific parameter sharing mechanisms [3], or training smaller models on outputs of larger models ("distillation") [11].

**Structured Matrices:** An $m \times n$ matrix which can be described in much fewer than $mn$ parameters is referred to as a *structured matrix*. Typically, the structure should not only reduce memory

requirements, but also dramatically accelerate inference and training via fast matrix-vector products and gradient computations. Below are classes of structured matrices arising pervasively in many contexts [18] with different types of parameter sharing (indicated by the color).

(i) *Toeplitz*

$$\begin{bmatrix} t_0 & t_{-1} & \cdots & t_{-(n-1)} \\ t_1 & t_0 & \cdots & \vdots \\ \vdots & \vdots & \vdots & t_{-1} \\ t_{n-1} & \cdots & t_1 & t_0 \end{bmatrix}$$

(ii) *Vandermonde*

$$\begin{bmatrix} 1 & v_0 & \cdots & v_0^{n-1} \\ 1 & v_1 & \cdots & v_1^{n-1} \\ \vdots & \vdots & \vdots & \vdots \\ 1 & v_{n-1} & \cdots & v_{n-1}^{n-1} \end{bmatrix}$$

(iii) *Cauchy*

$$\begin{bmatrix} \frac{1}{u_0-v_0} & \cdots & \cdots & \frac{1}{u_0-v_{n-1}} \\ \frac{1}{u_1-v_0} & \cdots & \cdots & \vdots \\ \vdots & \vdots & \vdots & \vdots \\ \frac{1}{u_{n-1}-v_0} & \cdots & \cdots & \frac{1}{u_{n-1}-v_{n-1}} \end{bmatrix}$$

*Toeplitz* matrices have constant values along each of their diagonals. When the same property holds for anti-diagonals, the resulting class of matrices are called *Hankel* matrices. Toeplitz and Hankel matrices are intimately related to one-dimensional discrete convolutions [10], and arise naturally in time series analysis and dynamical systems. A *Vandermonde* matrix is determined by taking elementwise powers of its second column. A very important special case is the complex matrix associated with the Discrete Fourier transform (DFT) which has Vandermonde structure with $v_j = \omega_n^j, j = 1 \ldots n$ where $\omega_n = \exp \frac{-2\pi i}{n}$ is the primitive $n^{th}$ root of unity. Similarly, the entries of $n \times n$ *Cauchy* matrices are completely defined by two length $n$ vectors. Vandermonde and Cauchy matrices arise naturally in polynomial and rational interpolation problems.

**"Superfast" Numerical Linear Algebra**: The structure in these matrices can be exploited for faster linear algebraic operations such as matrix-vector multiplication, inversion and factorization. In particular, the matrix-vector product can be computed in time $O(n \log n)$ for Toeplitz and Hankel matrices, and in time $O(n \log^2 n)$ for Vandermonde and Cauchy matrices.

**Displacement Operators**: At first glance, these matrices appear to have very different kinds of parameter sharing and consequently very different algorithms to support fast linear algebra. It turns out, however, that each structured matrix class described above, can be associated with a specific *displacement operator*, $L : \mathbb{R}^{m \times n} \mapsto \mathbb{R}^{m \times n}$ which transforms each matrix, say $\mathbf{M}$, in that class into an $m \times n$ matrix $L[\mathbf{M}]$ that has very low-rank, i.e. $rank(L[\mathbf{M}]) \ll \min(m, n)$. This displacement rank approach, which can be traced back to a seminal 1979 paper [13], greatly unifies algorithm design and complexity analysis for structured matrices [13], [18], [14].

**Generalizations of Structured Matrices:** Consider deriving a matrix by taking arbitrary linear combinations of products of structured matrices and their inverses, e.g. $\alpha_1 \mathbf{T}_1 \mathbf{T}_2^{-1} + \alpha_2 \mathbf{T}_3 \mathbf{T}_4^{-1} \mathbf{T}_5$ where each $\mathbf{T}_i$ is a Toeplitz matrix. The parameter sharing structure in such a derived matrix is by no means apparent anymore. Yet, it turns out that the associated displacement operator remarkably continues to expose the underlying parsimony structure, i.e. such derived matrices are still mapped to relatively low-rank matrices! The displacement rank approach allows fast linear algebra algorithms to be seamlessly extended to these broader classes of matrices. The displacement rank parameter controls the degree of structure in these generalized matrices.

**Technical Preview, Contributions and Outline**: We propose building deep learning pipelines where parameter matrices belong to the class of generalized structured matrices characterized by low displacement rank. In Section 2, we attempt to give a self-contained overview of the displacement rank approach [13], [18] drawing key results from the relevant literature on structured matrix computations (proved in our supplementary material [1] for completeness). In Section 3, we show that the proposed structured transforms for deep learning admit fast matrix multiplication and gradient computations, and have rich statistical modeling capacity that can be explicitly controlled by the displacement rank hyperparameter, covering, along a continuum, an entire spectrum of configurations from highly structured to unstructured matrices. While our focus in this paper is on Toeplitz-related transforms, our proposal extends to other structured matrix generalizations. In Section 4, we study inference and training-time acceleration with structured transforms as a function of displacement rank and dimensionality. We find that our approach compares highly favorably with numerous other techniques for learning size-constrained models on several benchmark datasets. Finally, we demonstrate our approach on mobile speech recognition applications where we are able to match the performance of much bigger state of the art models with a fraction of parameters.

**Notation**: Let $\mathbf{e}_1 \ldots \mathbf{e}_n$ denote the canonical basis elements of $\mathbb{R}^n$ (viewed as column vectors). $\mathbf{I}_n, \mathbf{0}_n$ denote $n \times n$ identity and zero matrices respectively. $\mathbf{J}_n = [\mathbf{e}_n \ldots \mathbf{e}_1]$ is the anti-identity reflection matrix whose action on a vector is to reverse its entries. When the dimension is obvious

we may drop the subscript; for rectangular matrices, we may specify both the dimensions explicitly, e.g. we use $0_{1 \times n}$ for a zero-valued row-vector, and $1_n$ for all ones column vector of length $n$. $\mathbf{u} \circ \mathbf{v}$ denotes Hadamard (elementwise) product between two vectors $\mathbf{v}, \mathbf{u}$. For a complex vector $\mathbf{u}, \bar{\mathbf{u}}$ will denote the vector of complex conjugate of its entries. The Discrete Fourier Transform (DFT) matrix will be denoted by $\Omega$ (or $\Omega_n$); we will also use $\mathbf{fft}(\mathbf{x})$ to denote $\Omega\mathbf{x}$, and $\mathbf{ifft}(\mathbf{x})$ to denote $\Omega^{-1}\mathbf{x}$. For a vector $\mathbf{v}$, $diag(\mathbf{v})$ denotes a diagonal matrix given by $diag(\mathbf{v})_{ii} = v_i$.

## 2 Displacement Operators associated with Structured Matrices

We begin by providing a brisk background on the displacement rank approach. Unless otherwise specified, for notational convenience we will henceforth assume squared transforms, i.e., $m = n$, and discuss rectangular transforms later. Proofs of various assertions can be found in our self-contained supplementary material [1] or in [18, 19].

The *Sylvester* displacement operator, denoted as $L = \nabla_{\mathbf{A},\mathbf{B}} : \mathbb{R}^{n \times n} \mapsto \mathbb{R}^{n \times n}$ is defined by,

$$\nabla_{\mathbf{A},\mathbf{B}}[\mathbf{M}] = \mathbf{AM} - \mathbf{MB} \qquad (1)$$

where $\mathbf{A} \in \mathbb{R}^{n \times n}, \mathbf{B} \in \mathbb{R}^{n \times n}$ are fixed matrices referred to as operator matrices. Closely related is the *Stein* displacement operator, denoted as $L = \triangle_{\mathbf{A},\mathbf{B}} : \mathbb{R}^{n \times n} \mapsto \mathbb{R}^{n \times n}$, and defined by,

$$\triangle_{\mathbf{A},\mathbf{B}}[\mathbf{M}] = \mathbf{M} - \mathbf{AMB} \qquad (2)$$

By carefully choosing $\mathbf{A}$ and $\mathbf{B}$ one can instantiate Sylvester and Stein displacement operators with desirable properties. In particular, for several important classes of displacement operators, $\mathbf{A}$ and/or $\mathbf{B}$ are chosen to be an $f$-**unit-circulant matrix** defined as follows.

**Definition 2.1** ($f$-unit-Circulant Matrix). *For a real-valued scalar $f$, the $(n \times n)$ $f$-circulant matrix, denoted by $\mathbf{Z}_f$, is defined as follows,*

$$\mathbf{Z}_f = [\mathbf{e}_2, \mathbf{e}_3 \dots \mathbf{e}_n, \ f\mathbf{e}_1] = \begin{bmatrix} 0 & 0 & \dots & f \\ 1 & 0 & \dots & 0 \\ \vdots & \vdots & \vdots & \vdots \\ 0 & \dots & 1 & 0 \end{bmatrix} = \begin{bmatrix} 0_{1 \times (n-1)} & f \\ \mathbf{I}_{n-1} & 0_{(n-1) \times 1} \end{bmatrix}$$

The $f$-unit-circulant matrix is associated with a basic *downward shift-and-scale* transformation, i.e., the matrix-vector product $\mathbf{Z}_f\mathbf{v}$ shifts the elements of the column vector $\mathbf{v}$ "downwards", and scales and brings the last element $v_n$ to the "top", resulting in $[fv_n, v_1, \dots v_{n-1}]^T$. It has several basic algebraic properties (see Proposition 1.1 [1]) that are crucial for the results stated in this section

Figure 1 lists the rank of the Sylvester displacement operator in Eqn 1 when applied to matrices belonging to various structured matrix classes, where the operator matrices $\mathbf{A}, \mathbf{B}$ in Eqn. 1 are chosen to be diagonal and/or $f$-unit-circulant. It can be seen that despite the difference in their structures, all these classes are characterized by very low displacement rank. Figure 2 shows how this low-rank transformation happens in the case of a $4 \times 4$ Toeplitz matrix (also see section 1, Lemma 1.2 [1]). Embedded in the $4 \times 4$ Toeplitz matrix $\mathbf{T}$ are two copies of a $3 \times 3$ Toeplitz matrix shown in black and red boxes. The shift and scale action of $\mathbf{Z}_1$ and $\mathbf{Z}_{-1}$ aligns these sub-matrices. By taking the difference, the Sylvester displacement operator nullifies the aligned submatrix leaving a rank 2 matrix with non-zero elements only along its first row and last column. Note that the negative sign introduced by $\mathbf{TZ}_{-1}$ term prevents the complete zeroing out of the value of $t$ (marked by red star) and is hence critical for invertibility of the displacement action.

Figure 1: Below r is $rank(\nabla_{\mathbf{A},\mathbf{B}}[\mathbf{M}])$

| Structured Matrix $\mathbf{M}$ | $\mathbf{A}$ | $\mathbf{B}$ | $r$ |
|---|---|---|---|
| Toeplitz $\mathbf{T}, \mathbf{T}^{-1}$ | $\mathbf{Z}_1$ | $\mathbf{Z}_{-1}$ | $\leq 2$ |
| Hankel $\mathbf{H}, \mathbf{H}^{-1}$ | $\mathbf{Z}_1$ | $\mathbf{Z}_0^T$ | $\leq 2$ |
| $\mathbf{T} + \mathbf{H}$ | $\mathbf{Z}_0 + \mathbf{Z}_0^T$ | $\mathbf{Z}_0 + \mathbf{Z}_0^T$ | $\leq 4$ |
| Vandermonde $V(\mathbf{v})$ | $diag(\mathbf{v})$ | $\mathbf{Z}_0$ | $\leq 1$ |
| $V(\mathbf{v})^{-1}$ | $\mathbf{Z}_0$ | $diag(\mathbf{v})$ | $\leq 1$ |
| $V(\mathbf{v})^T$ | $\mathbf{Z}_0^T$ | $diag(\mathbf{v})$ | $\leq 1$ |
| Cauchy $\mathbf{C}(\mathbf{s},\mathbf{t})$ | $diag(\mathbf{s})$ | $diag(\mathbf{t})$ | $\leq 1$ |
| $\mathbf{C}(\mathbf{s},\mathbf{t})^{-1}$ | $diag(\mathbf{t})$ | $diag(\mathbf{s})$ | $\leq 1$ |

Figure 2: Displacement Action on Toeplitz Matrix

Each class of structured matrices listed in Figure 1 can be naturally generalized by allowing the rank of the displacement operator to be higher. Specifically, given a displacement operator $L$, and displacement rank parameter $r$, one may consider the class of matrices $\mathbf{M}$ that satisfies $rank(L(\mathbf{M})) \leq r$. Clearly then, $L[\mathbf{M}] = \mathbf{G}\mathbf{H}^T$ for rank $r$ matrices $\mathbf{G}, \mathbf{H}$. We refer to $rank(L(\mathbf{M}))$ as the *displacement rank* of $\mathbf{M}$ under $L$, and to the low-rank factors $\mathbf{G}, \mathbf{H} \in \mathbb{R}^{n \times r}$ as the associated *low-displacement generators*. For the operators listed in Table 1, these broader classes of structured matrices are correspondingly called *Toeplitz-like*, *Vandermonde-like* and *Cauchy-like*. Fast numerical linear algebra algorithms extend to such matrices [18].

In order to express structured matrices with low-displacement rank directly as a function of its low-displacement generators, we need to invert $L$ and obtain a learnable parameterization. For Stein type displacement operator, the following elegant result is known (see proof in [1]):

**Theorem 2.2** ( [19], Krylov Decomposition). *If an $n \times n$ matrix $\mathbf{M}$ is such that $\triangle_{\mathbf{A},\mathbf{B}}[\mathbf{M}] = \mathbf{G}\mathbf{H}^T$ where $\mathbf{G} = [\mathbf{g}_1 \dots \mathbf{g}_r], \mathbf{H} = [\mathbf{h}_1 \dots \mathbf{h}_r] \in \mathbb{R}^{n \times r}$ and the operator matrices satisfy: $\mathbf{A}^n = a\mathbf{I}$, $B^n = b\mathbf{I}$ for some scalars $a, b$, then $\mathbf{M}$ can be expressed as:*

$$\mathbf{M} = \frac{1}{1-ab} \sum_{j=1}^{r} krylov(\mathbf{A}, \mathbf{g}_j) krylov(\mathbf{B}^T, \mathbf{h}_j)^T \tag{3}$$

*where $krylov(\mathbf{A}, \mathbf{v})$ is defined by:*

$$krylov(\mathbf{A}, \mathbf{v}) = [\mathbf{v} \ \ \mathbf{A}\mathbf{v} \ \ \mathbf{A}^2\mathbf{v} \dots \mathbf{A}^{n-1}\mathbf{v}] \tag{4}$$

Henceforth, our focus in this paper will be on Toeplitz-like matrices for which the displacement operator of interest (see Table 1) is of Sylvester type: $\nabla_{\mathbf{Z}_1,\mathbf{Z}_{-1}}$. In order to apply Theorem 2.2, one can switch between Sylvester and Stein operators, setting $\mathbf{A} = \mathbf{Z}_1$ and $\mathbf{B} = \mathbf{Z}_{-1}$ which both satisfy the conditions of Theorem 2.2 (see property 3, Proposition 1.1 [1]). The resulting expressions involve Krylov matrices generated by $f$-*unit-circulant matrices* which are called $f$-*circulant matrices* in the literature.

**Definition 2.3** ($f$-circulant matrix). *Given a vector $\mathbf{v}$, the f-Circulant matrix, $\mathbf{Z}_f(\mathbf{v})$, is defined as follows:*

$$\mathbf{Z}_f(\mathbf{v}) = krylov(\mathbf{Z}_f, \mathbf{v}) = \begin{bmatrix} v_0 & fv_{n-1} & \dots & fv_1 \\ v_1 & v_0 & \dots & fv_2 \\ \vdots & \vdots & \vdots & fv_{n-1} \\ v_{n-1} & \dots & v_1 & v_0 \end{bmatrix}$$

*Two special cases are of interest: $f = 1$ corresponds to* **Circulant** *matrices, and $f = -1$ corresponds to* **skew-Circulant** *matrices.*

Finally, one can obtain an explicit parameterization for Toeplitz-like matrices which turns out to involve taking sums of products of Circulant and skew-Circulant matrices.

**Theorem 2.4** ([18]). *If an $n \times n$ matrix $\mathbf{M}$ satisfies $\nabla_{\mathbf{Z}_1,\mathbf{Z}_{-1}}[\mathbf{M}] = \mathbf{G}\mathbf{H}^T$ where $\mathbf{G} = [\mathbf{g}_1 \dots \mathbf{g}_r], \mathbf{H} = [\mathbf{h}_1 \dots \mathbf{h}_r] \in \mathbb{R}^{n \times r}$, then $\mathbf{M}$ can be written as:*

$$\mathbf{M} = \frac{1}{2} \sum_{j=1}^{r} \mathbf{Z}_1(\mathbf{g}_j) \mathbf{Z}_{-1}(\mathbf{J}\mathbf{h}_j) \tag{5}$$

## 3 Learning Toeplitz-like Structured Transforms

Motivated by Theorem 2.4, we propose learning parameter matrices of the form in Eqn. 5 by optimizing the displacement factors $\mathbf{G}, \mathbf{H}$. First, from the properties of displacement operators [18], it follows that this class of matrices is very rich from a statistical modeling perspective.

**Theorem 3.1** (Richness). *The set of all $n \times n$ matrices that can be written as,*

$$\mathbf{M}(\mathbf{G}, \mathbf{H}) = \sum_{i=1}^{r} \mathbf{Z}_1(\mathbf{g}_i) \mathbf{Z}_{-1}(\mathbf{h}_i) \tag{6}$$

*for some $\mathbf{G} = [\mathbf{g}_1 \dots \mathbf{g}_r], \mathbf{H} = [\mathbf{h}_1 \dots \mathbf{h}_r] \in \mathbb{R}^{n \times r}$ contains:*

- *All $n \times n$ Circulant and Skew-Circulant matrices for $r \geq 1$.*
- *All $n \times n$ Toeplitz matrices for $r \geq 2$.*
- *Inverses of Toeplitz matrices for $r \geq 2$.*
- *All products of the form $\mathbf{A}_1 \ldots \mathbf{A}_t$ for $r \geq 2t$.*
- *All linear combinations of the form $\sum_{i=1}^p \beta_i \mathbf{A}_1^{(i)} \ldots \mathbf{A}_t^{(i)}$ where $r \geq 2tp$.*
- *All $n \times n$ matrices for $r = n$.*

*where each $\mathbf{A}_i$ above is a Toeplitz matrix or the inverse of a Toeplitz matrix.*

When we learn a parameter matrix structured as Eqn. 6 with displacement rank equal to 1 or 2, we also search over convolutional transforms. In this sense, structured transforms with higher displacement rank generalize (one-dimensional) convolutional layers. The displacement rank provides a knob on modeling capacity: low displacement matrices are highly structured and compact, while high displacement matrices start to contain increasingly unstructured dense matrices.

Next, we show that associated structured transforms of the form $f(\mathbf{x}) = \mathbf{M}(\mathbf{G}, \mathbf{H})\mathbf{x}$ admit fast evaluation, and gradient computations with respect to $\mathbf{G}, \mathbf{H}$. First we recall the following well-known result concerning the diagonalization of $f$-Circulant matrices.

**Theorem 3.2** (Diagonalization of $f$-circulant matrices, Theorem 2.6.4 [18]). *For any $f \neq 0$, let $\mathbf{f} = [1, f^{\frac{1}{n}}, f^{\frac{2}{n}}, \ldots f^{\frac{n-1}{n}}]^T \in \mathbb{C}^n$, and $\mathbf{D_f} = diag(\mathbf{f})$. Then,*

$$\mathbf{Z}_f(\mathbf{v}) = \mathbf{D_f}^{-1} \, \Omega^{-1} \, diag(\Omega(\mathbf{f} \circ \mathbf{v}))\Omega\mathbf{D_f} \tag{7}$$

This result implies that for the special cases of $f = 1$ and $f = -1$ corresponding to Circulant and Skew-circulant matrices respectively, the matrix-vector multiplication can be computed in $O(n \log n)$ time via the Fast Fourier transform:

$$\mathbf{y} = \mathbf{Z}_1(\mathbf{v})\mathbf{x} \quad = \quad \mathbf{ifft}\left(\mathbf{fft}(\mathbf{v}) \circ \mathbf{fft}(\mathbf{x})\right) \tag{8}$$

$$\mathbf{y} = \mathbf{Z}_1(\mathbf{v})^T\mathbf{x} \quad = \quad \mathbf{ifft}\left(\overline{\mathbf{fft}(\mathbf{v})} \circ \mathbf{fft}(\mathbf{x})\right) \tag{9}$$

$$\mathbf{y} = \mathbf{Z}_{-1}(\mathbf{v})\mathbf{x} \quad = \quad \bar{\boldsymbol{\eta}} \circ \mathbf{ifft}\left(\mathbf{fft}(\boldsymbol{\eta} \circ \mathbf{v}) \circ \mathbf{fft}(\boldsymbol{\eta} \circ \mathbf{x})\right) \tag{10}$$

$$\mathbf{y} = \mathbf{Z}_{-1}(\mathbf{v})^T\mathbf{x} \quad = \quad \bar{\boldsymbol{\eta}} \circ \mathbf{ifft}\left(\mathbf{fft}(\overline{\boldsymbol{\eta} \circ \mathbf{v}}) \circ \mathbf{fft}(\boldsymbol{\eta} \circ \mathbf{x})\right) \tag{11}$$

where $\boldsymbol{\eta} = [1, \eta, \eta^2 \ldots \eta^{n-1}]^T$ where $\eta = (-1)^{\frac{1}{n}} = \exp(i\frac{\pi}{n})$, the root of negative unity.

In particular, a single matrix-vector product for Circulant and Skew-circulant matrices has the computational cost of 3 FFTs. Therefore, for matrices of the form in Eqn. 6 comprising of $r$ products of Circulant and Skew-Circulant matrices, naively computing a matrix-vector product for a batch of $b$ input vectors would take $6rb$ FFTs. However, this cost can be significantly lowered to that of $2(rb + r + b)$ FFTs by making the following observation:

$$\mathbf{Y} = \sum_{i=1}^r \mathbf{Z}_1(\mathbf{g}_i)\mathbf{Z}_{-1}(\mathbf{h}_i)\mathbf{X} = \Omega^{-1}\left(\sum_{i=1}^r \, diag(\Omega\mathbf{g}_i) \, \Omega \, diag(\bar{\boldsymbol{\eta}}) \, \Omega^{-1} \, diag(\Omega(\boldsymbol{\eta} \circ \mathbf{h}_i))\tilde{\mathbf{X}}\right)$$

where $\tilde{\mathbf{X}} = \Omega \, diag(\boldsymbol{\eta}) \, \mathbf{X}$. Here, (1) The FFT of the parameters, $\Omega\mathbf{g}_i$ and $\Omega(\boldsymbol{\eta} \circ \mathbf{h}_i)$ is computed once and shared across multiple input vectors in the minibatch, (2) The (scaled) FFT of the input, $(\Omega \, diag(\boldsymbol{\eta}) \, \mathbf{X})$ is computed once and shared across the sum in Eqn. 6, and (3) The final inverse FFT is also shared. Thus, the following result is immediate.

**Theorem 3.3** (Fast Multiplication). *Given an $n \times b$ matrix $\mathbf{X}$, the matrix-matrix product, $\mathbf{Y} = \left(\sum_{i=1}^r \mathbf{Z}_1(\mathbf{g}_i)\mathbf{Z}_{-1}(\mathbf{h}_i)\right) \mathbf{X}$, can be computed at the cost of $2(rb + b + r)$ FFTs, using the following algorithm.*

▶ *Set $\boldsymbol{\eta} = [1, \eta, \eta^2 \ldots \eta^{n-1}]^T$ where $\eta = (-1)^{\frac{1}{n}} = \exp(i\frac{\pi}{n})$*
▶ *Initialize $\mathbf{Y} = 0_{n \times b}$*
▶ *Set $\tilde{\mathbf{X}} = \mathbf{fft}(\mathbf{diag}(\boldsymbol{\eta})\mathbf{X})$*
▶ *Set $\tilde{\mathbf{G}} = \mathbf{fft}(\mathbf{G}) = [\tilde{\mathbf{g}}_1 \ldots \tilde{\mathbf{g}}_r]$ and $\tilde{\mathbf{H}} = \mathbf{fft}(\mathbf{diag}(\boldsymbol{\eta})\mathbf{H}) = [\tilde{\mathbf{h}}_1 \ldots \tilde{\mathbf{h}}_r]$*
▶ *for i = 1 to r*
  ○ *$\mathbf{U} = \mathbf{Z}_{-1}(\mathbf{h}_i)\mathbf{X} = diag(\bar{\boldsymbol{\eta}})\mathbf{ifft}\left(\mathbf{diag}(\tilde{\mathbf{h}}_i)\tilde{\mathbf{X}}\right)$*
  ○ *$\mathbf{V} = diag(\tilde{\mathbf{g}}_i) \, \mathbf{fft}(\mathbf{U})$*

$\circ$ $\mathbf{Y} = \mathbf{Y} + \mathbf{V}$
► *Set* $\mathbf{Y} = \mathbf{ifft}\,(\mathbf{Y})$
► *Return* $\mathbf{Y}$

We now show that when our structured transforms are embedded in a deep learning pipeline, the gradient computation can also be accelerated. First, we note that the Jacobian structure of $f$-Circulant matrices has the following pleasing form.

**Proposition 3.4** (Jacobian of $f$-circulant transforms). *The Jacobian of the map $f(\mathbf{x}, \mathbf{v}) = \mathbf{Z}_f(\mathbf{v})\mathbf{x}$ with respect to the parameters $\mathbf{v}$ is $\mathbf{Z}_f(\mathbf{x})$.*

This leads to the following expressions for the Jacobians of the structured transforms of interest.

**Proposition 3.5** (Jacobians with respect to displacement generators $\mathbf{G}, \mathbf{H}$). *Consider parameterized vector-valued transforms of the form,*

$$f(\mathbf{x}, \mathbf{G}, \mathbf{H}) = \sum_{i=1}^{r} \mathbf{Z}_1(\mathbf{g}_i)\mathbf{Z}_{-1}(\mathbf{h}_i)\mathbf{x} \tag{12}$$

*The Jacobians of $f$ with respect to the $j^{th}$ column of $\mathbf{G}, \mathbf{H}$, i.e. $\mathbf{g}_j, \mathbf{h}_j$, at $\mathbf{x}$, are as follows:*

$$\begin{aligned} J_{\mathbf{g}_j} f|_{\mathbf{x}} &= \mathbf{Z}_1\left(\mathbf{Z}_{-1}(\mathbf{h}_j)\mathbf{x}\right) \tag{13} \\ J_{\mathbf{h}_j} f|_{\mathbf{x}} &= \mathbf{Z}_1(\mathbf{g}_j)\mathbf{Z}_{-1}(\mathbf{x}) \tag{14} \end{aligned}$$

Based on Eqns. 13, 14 the gradient over a minibatch of size $b$ requires computing, $\sum_i^b [J_{\mathbf{g}_j} f|_{\mathbf{x_i}}]^T \boldsymbol{\delta}_i$ and $\sum_{i=1}^b [J_{\mathbf{h}_j} f|_{\mathbf{x_i}}]^T \boldsymbol{\delta}_i$ where, $\{\mathbf{x}_i\}_{i=1}^b$ and $\{\boldsymbol{\delta}_i\}_{i=1}^b$ are batches of forward and backward inputs during backpropagation. These can be naively computed with $6rb$ FFTs. However, as before, by sharing FFT of the forward and backward inputs, and the fft of the parameters, this can be lowered to $(4br + 4r + 2b)$ FFTs. Below we give matricized implementation.

**Proposition 3.6** (Fast Gradients). *Let $\mathbf{X}, \mathbf{Z}$ be $n \times b$ matrices whose columns are forward and backward inputs respectively of minibatch size $b$ during backpropagation. The gradient with respect to $\mathbf{g}_j, \mathbf{h}_j$ can be computed at the cost of $(4br + 4r + 2b)$ FFTs as follows:*

► *Compute* $\tilde{\mathbf{Z}} = \mathbf{fft}(\mathbf{Z}), \tilde{\mathbf{X}} = \mathbf{fft}(\mathrm{diag}(\boldsymbol{\eta})\mathbf{X}), \tilde{\mathbf{G}} = \mathbf{fft}(\mathbf{G}), \tilde{\mathbf{H}} = \mathbf{fft}(\mathrm{diag}(\boldsymbol{\eta})\mathbf{H})$
► *Gradient wrt* $\mathbf{g}_j$ *($2b+1$ FFTs)*
  $\circ$ *return* $\mathbf{ifft}\left[\left(\overline{\mathbf{fft}\left(\mathrm{diag}(\bar{\boldsymbol{\eta}})\mathbf{ifft}\left(\mathrm{diag}(\tilde{\mathbf{h}}_\mathbf{j})\tilde{\mathbf{X}}\right)\right)} \circ \tilde{\mathbf{Z}}\right)\mathbf{1_b}\right]$
► *Gradient wrt* $\mathbf{h}_j$ *($2b+1$ FFTs)*
  $\circ$ *return* $\mathrm{diag}\left(\bar{\boldsymbol{\eta}}\right)\mathbf{ifft}\left[\left(\overline{\tilde{\mathbf{X}}} \circ \mathbf{fft}\left(\mathrm{diag}(\boldsymbol{\eta})\mathbf{ifft}\left(\mathrm{diag}(\bar{\tilde{\mathbf{g}}}_\mathbf{i})\tilde{\mathbf{Z}}\right)\right)\right)\mathbf{1_b}\right]$

**Rectangular Transforms**: Variants of Theorems 2.2, 2.4 exist for rectangular transforms, see [19]. Alternatively, for $m < n$ we can subsample the outputs of square $n \times n$ transforms at the cost of extra computations, while for $m > n$, assuming $m$ is a multiple of $n$, we can stack $\frac{m}{n}$ output vectors of square $n \times n$ transforms.

# 4 Empirical Studies

**Acceleration with Structured Transforms**: In Figure 3, we analyze the speedup obtained in practice using $n \times n$ Circulant and Toeplitz-like matrices relative to a dense unstructured $n \times n$ matrix (fully connected layer) as a function of displacement rank and dimension $n$. Three scenarios are considered: inference speed per test instance, training speed as implicitly dictated by forward passes on a minibatch, and gradient computations on a minibatch. Factors such as differences in cache optimization, SIMD vectorization and multithreading between Level-2 BLAS (matrix-vector multiplication), Level-3 BLAS (matrix-matrix multiplication) and FFT implementations (we use FFTW: http://www.fftw.org) influence the speedup observed in practice. Speedup gains start to show for dimensions as small as $512$ for Circulant matrices. The gains become dramatic with acceleration of the order of 10 to 100 times for several thousand dimensions, even for higher displacement rank Toeplitz-like transforms.

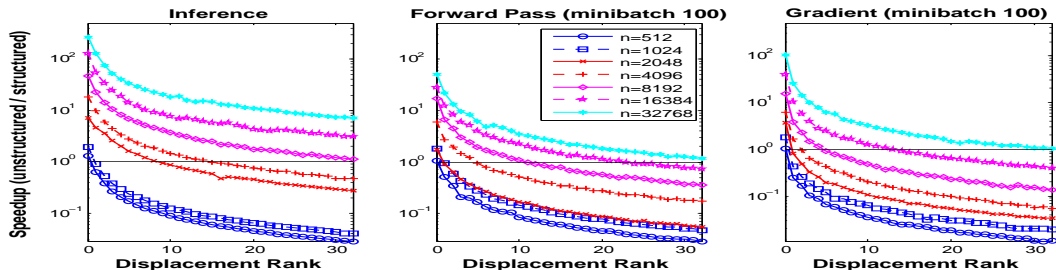

Figure 3: Acceleration with $n \times n$ Structured Transforms (6-core 32-GB Intel(R) Xeon(R) machine; random datasets). In the plot, displacement rank = 0 corresponds to a Circulant Transform.

**Effectiveness for learning compact Neural Networks**: Next, we compare the proposed structured transforms with several existing techniques for learning compact feedforward neural networks. We exactly replicate the experimental setting from the recent paper on HASHEDNETS [3] which uses several image classification datasets first prepared by [15]. MNIST is the original 10-class MNIST digit classification dataset with 60000 training examples and 10000 test examples. BG-IMG-ROT refers to a challenging version of MNIST where digits are randomly rotated and placed against a random black and white background. RECT (1200 training images, 50000 test images) and CONVEX (8000 training images, 50000 test images) are 2-class binary image datasets where the task is to distinguish between tall and wide rectangles, and whether the "on" pixels form a convex region or not, respectively. In all datasets, input images are of size $28 \times 28$. Several existing techniques are benchmarked in [3] for compressing a reference single hidden layer model with 1000 hidden nodes.

- Random Edge Removal (RER) [5] where a fraction of weights are randomly frozen to be zero-valued.
- Low-rank Decomposition (LRD) [9]
- Neural Network (NN) where the hidden layer size is reduced to satisfy a parameter budget.
- Dark Knowledge (DK) [11]: A small neural network is trained with respect to both the original labeled data, as well as soft targets generated by a full uncompressed neural network.
- HashedNets (HN) [3]: This approach uses a low-cost hash function to randomly group connection weights which share the same value.
- HashedNets with Dark Knowledge ($HN^{DK}$): Trains a HashedNet with respect to both the original labeled data, as well as soft targets generated by a full uncompressed neural network.

We consider learning models of comparable size with the weights in the hidden layer structured as a Toeplitz-like matrix. We also compare with the FASTFOOD approach of [25, 16] where the weight matrix is a product of diagonal parameter matrices and fixed permutation and Walsh-Hadamard matrices, also admitting $O(n \log n)$ multiplication and gradient computation time. The CIRCULANT Neural Network approach proposed in [4] is a special case of our framework (Theorem 3.1).

Results in Table 1 show that Toeplitz-like structured transforms outperform all competing approaches on all datasets, sometimes by a very significant margin, with similar or drastically lesser number of parameters. It should also be noted that while random weight tying in HASHEDNETS reduces the number of parameters, the lack of structure in the resulting weight matrix cannot be exploited for FFT-like $O(n \log n)$ multiplication time. We note in passing that for HASHEDNETS weight matrices whose entries assume only one of $B$ distinct values, the Mailman algorithm [17] can be used for faster matrix-vector multiplication, with complexity $O(n^2 \log(B)/(\log n))$, which still is much slower than matrix-vector multiplication time for Toeplitz-like matrices. Also note that the distillation ideas of [11] are complementary to our approach and can further improve our results.

| | RER | LRD | NN | DK | HN | $HN^{DK}$ | Fastfood | CIRCULANT | TOEPLITZ (1) | TOEPLITZ (2) | TOEPLITZ (3) |
|---|---|---|---|---|---|---|---|---|---|---|---|
| MNIST | 15.03 | 28.99 | 6.28 | 6.32 | 2.79 | 2.65 | 6.61 | 3.12 | 2.79 | 2.54 | **2.09** |
| | *12406* | *12406* | *12406* | *12406* | *12406* | *12406* | *10202* | *8634* | *9418* | *10986* | *12554* |
| BG-IMG-ROT | 73.17 | 80.63 | 79.03 | 77.40 | 59.20 | 58.25 | 68.4 | 62.11 | 57.66 | 55.21 | **53.94** |
| | *12406* | *12406* | *12406* | *12406* | *12406* | *12406* | *10202* | *8634* | *9418* | *10986* | *12554* |
| CONVEX | 37.22 | 39.93 | 34.37 | 31.85 | 31.77 | 30.43 | 33.92 | 24.76 | 17.43 | **16.18** | 20.23 |
| | *12281* | *12281* | *12281* | *12281* | *12281* | *12281* | *3922* | *2352* | *3138* | *4706* | *6774* |
| RECT | 18.23 | 23.67 | 5.68 | 5.78 | 3.67 | 3.37 | 21.45 | 2.91 | 0.70 | 0.89 | **0.66** |
| | *12281* | *12281* | *12281* | *12281* | *12281* | *12281* | *3922* | *2352* | *3138* | *4706* | *6774* |

Table 1: Error rate and number of parameters (italicized). Best results in **blue**.

**Mobile Speech Recognition**: We now demonstrate the techniques developed in this paper on a speech recognition application meant for mobile deployment. Specifically, we consider a keyword spotting (KWS) task, where a deep neural network is trained to detect a specific phrase, such as "Ok Google" [2]. The data used for these experiments consists of $10-15K$ utterances of selected phrases (such as "play-music", "decline-call"), and a larger set of $396K$ utterances to serve as negative training examples. The utterances were randomly split into training, development and evaluation sets in the ratio of $80:5:15$. We created a noisy evaluation set by artificially adding babble-type cafeteria noise at 0dB SNR to the "play-music" clean data set. We will refer to this noisy data set as CAFE0. We refer the reader to [23] for more details about the datasets. We consider the task of shrinking a large model for this task whose architecture is as follows [23]: the input layer consists of 40 dimensional log-mel filterbanks, stacked with a temporal context of 32, to produce an input of $32 \times 40$ whose dimensions are in time and frequency respectively. This input is fed to a convolutional layer with filter size $32 \times 8$, frequency stride 4 and 186 filters. The output of the convolutional layer is of size $9 \times 186 = 1674$. The output of this layer is fed to a $1674 \times 1674$ fully connected layer, followed by a softmax layer for predicting 4 classes constituting the phrase "play-music". The full training set contains about 90 million samples. We use asynchronous distributed stochastic gradient descent (SGD) in a parameter server framework [8], with 25 worker nodes for optimizing various models. The global learning rate is set to $0.002$, while our structured transform layers use a layer-specific learning rate of $0.0005$; both are decayed by an exponential factor of $0.1$.

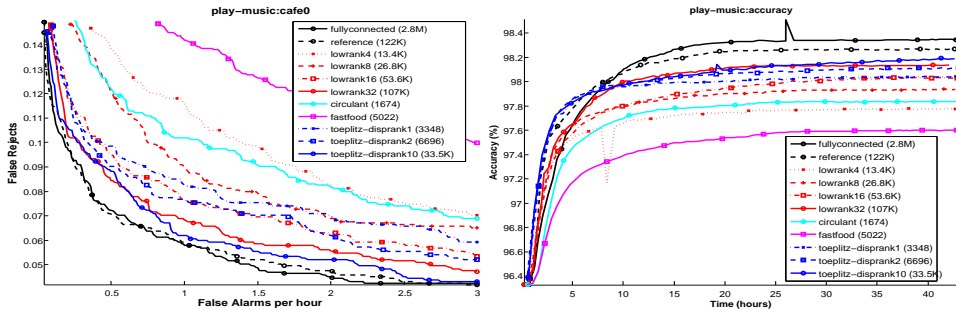

Figure 4: "play-music" detection performance: (left) End-to-end keyword spotting performance in terms of false reject (FR) rate per false alarm (FA) rate (lower is better) (right): Classification accuracy as a function of training time. Displacement rank is in parenthesis for Toeplitz-like models.

Results with 11 different models are reported in Figure 4 (left) including the state of the art keyword spotting model developed in [23]. At an operating point of 1 False Alarm per hour, the following observations can be made: With just 3348 parameters, a displacement rank=1 TOEPLITZ-LIKE structured transform outperforms a standard low-rank bottleneck model with rank=16 containing 16 times more parameters; it also lowers false reject rates from $10.2\%$ with CIRCULANT and $14.2\%$ with FASTFOOD transforms to about $8.2\%$. With displacement rank 10, the false reject rate is $6.2\%$, in comparison to $6.8\%$ with the 3 times larger rank=32 standard low-rank bottleneck model. Our best Toeplitz-like model comes within $0.4\%$ of the performance of the 80-times larger fully-connected and 3.6 times larger reference [23] models. In terms of raw classification accuracy as a function of training time, Figure 4 (right) shows that our models (with displacement ranks $1, 2$ and $10$) come within $0.2\%$ accuracy of the fully-connected and reference models, and easily provide much better accuracy-time tradeoffs in comparison to standard low-rank bottleneck models, Circulant and Fastfood baselines. The conclusions are similar for other noise conditions (see supplementary material [1]).

## 5 Perspective

We have introduced and shown the effectiveness of new notions of parsimony rooted in the theory of structured matrices. Our proposal can be extended to various other structured matrix classes, including Block and multi-level Toeplitz-like [12] matrices related to multidimensional convolution [21]. We hope that such ideas might lead to new generalizations of Convolutional Neural Networks.

**Acknowledgements**: We thank Yu-hsin Chen, Carolina Parada, Rohit Prabhavalkar, Alex Gruenstein, Rajat Monga, Baris Sumengen, Kilian Weinberger and Wenlin Chen for their contributions.

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
