[Reviews · NeurIPS 2015]

Submitted by Assigned_Reviewer_1

In this paper, the author proposes a method for fast matrix-vector multiplication and gradient computation for structured matrices. The notion of structured matrices is a generalized version of the Toeplitz matrix. Nevertheless, the technique of this paper can also be generalized to capture Vandermonde and Cauchy matrices.

The main idea behind the methodology is to represent the matrix as a product of Z_1(G) and Z_{-1}(H), where both G and H are tall matrices. This is a generalization of the low-rank factorization. The author shows that this new representation not only captures the low-rank matrices, but also the product, inverse and linear combination of Toeplitz matrices. It was also shown that the matrix-vector product can be computed efficiently through a small number of FFTs, so that it is order-of-magnitude faster than dense matrix-vector multiplication.

The idea is clever and is promising in accelerating the training of neural networks. It is also useful for compressing the representation of a neural network on mobile devices.

Here are some comment:

1. Toeplitz is closely related to one-dimensional convolution, but in practice use the convolution is usually two-dimensional. It means that using the Toeplitz representation may lose some information captured by the ordinary CNN. To see the impact of this loss, it is important to include ordinary CNN as a baseline algorithm, and compare algorithms on datasets other than MNIST. Although one dimensional convolution may be sufficient for MNIST classification. It may not be the case for natural images.

2. It is somewhat surprising that when n = 512, the implementation proposed in this paper is almost surely slower than the dense matrix multiplication. The author may want to discuss a little why it happens. It seems that the BLAS implementation of dense matrix-vector multiplication is 100x faster for executing task of the same computation complexity?

3. In mobile search recognition, what is the specific algorithm for training these models? Does the author train a fully connected network at first, then shrink it to be a smaller one, or does the author train the smaller model directly? If it was the first case, then what is the algorithm for shrinking the network?
Summary: In this paper, the author proposes a method for fast matrix-vector multiplication and gradient computation for structured matrices. The idea is clever and is promising in accelerating the training of neural networks. It is also useful for compressing the representation of a neural network on mobile devices.

Submitted by Assigned_Reviewer_2

The paper aims to cast a very wide net covering structured matrices and displacement operators before moving to what exactly has been investigated and shown in the paper which is the class of matrices proposed in section 3 and used in the experiments. The authors do not discuss their approach until page 5, where they finally propose using structured Toeplitz matrices and show how to efficiently compute the action of such a layer and its derivatives. The authors performed very extensive experiments comparing with various techniques for reducing the resources used by a neural network. Empirically structured toeplitz matrices seem to work better than other techniques at a given budget. However, I would have liked to see a plot

of a Pareto frontier between resources (# parameters) and accuracy, where every method is a curve (since all of the suggested methods have a knob for adjusting the number of parameters). This way we can compare the various methods across multiple budgets.

Typos:

line 192: v_2 -> f v_2 line 362: lesser number of parameters -> fewer parameters
Summary: Low-rank structure and convolutions have been by far the most popular ways to reduce capacity and resources in (deep) learning.

This paper proposes and experiments with Toeplitz-like structured transforms which seem to work very well in practice for replacing unstructured fully connected layers at the cost of a few FFTs.

Submitted by Assigned_Reviewer_3

Quality:

It is a little disappointing that Figure 2 shows that, if one is willing to spend just a few more hours, one can get better accuracy by using a fully-connected network. Larger model sizes (e.g. convolutional nets, where those few extra training hours may easily turn into a few extra days) might have better showcased the advantage of the proposed parsimonious models. Nevertheless, the advantage over other parsimonious models is certainly apparent on the speech recognition applications in the evaluation.

Clarity:

The paper could have used a bit more introductory material to set the stage (especially for readers who are not so familiar with linear algebra).

A bigger issue is that the first half of the paper is presented like a primer on linear algebra, and it takes a while to explain that the whole point is to learn matrices M with a certain structure. I would prefer if this point had been brought up earlier on. I have to say that the current presentation almost struck me as bad taste, as if the intent was to make the core idea seem more complicated than it really is.

Originality:

To the best of my knowledge, this is the first time I am seeing such low-rank transforms in the context of a Machine Learning application.

Significance:

Storage and computational costs for deep learning are a significant obstacle to their wider application. Work that dramatically reduces these costs, and thus makes them more accessible to the wider ML community, is of clear value.

I have read the author feedback and my decision remains unchanged.
Summary: The paper proposes fairly general low-rank parameterizations for structured transforms f() (e.g. in deep learning). The idea is technically attractive and seems like it can be easily retrofitted to existing ML programs or distributed execution frameworks.

Empirical results show that the proposed parameterizations close (but do not eliminate) the gap with fully connected neural networks, versus other parsimonious parameterizations. Overall, I think the idea deserves to be presented to the NIPS community, despite a few flaws in the paper.

Author Feedback
Author rebuttal: We thank the reviewers for their thoughtful comments and encouraging feedback. We will incorporate the suggestions made in the reviews.

To Assigned_Reviewer_4: We use asynchronous distributed stochastic gradient descent (SGD) in a parameter server framework [7], with 25 worker nodes for optimizing various models. The hyper-parameters are reported in the paper. The smaller models are trained directly.